# PARG Protein Regulation Roles in *Drosophila* Longevity Control

**DOI:** 10.3390/ijms25116189

**Published:** 2024-06-04

**Authors:** Guillaume Bordet, Alexei V. Tulin

**Affiliations:** School of Medicine and Health Sciences, Department of Biomedical Sciences, University of North Dakota, Grand Forks, ND 58202, USA; guillaume.bordet@und.edu

**Keywords:** PARG, PARP1, *Drosophila melanogaster*, lifespan, aging processes

## Abstract

Aging, marked by a gradual decline in physiological function and heightened vulnerability to age-related diseases, remains a complex biological process with multifaceted regulatory mechanisms. Our study elucidates the critical role of poly(ADP–ribose) glycohydrolase (PARG), responsible for catabolizing poly(ADP–ribose) (pADPr) in the aging process by modulating the expression of age-related genes in *Drosophila melanogaster*. Specifically, we uncover the regulatory function of the uncharacterized PARG C-terminal domain in controlling PARG activity. Flies lacking this domain exhibit a significantly reduced lifespan compared to wild-type counterparts. Furthermore, we observe progressive dysregulation of age-related gene expression during aging, accelerated in the absence of PARG activity, culminating in a premature aging phenotype. Our findings reveal the critical involvement of the pADPr pathway as a key player in the aging process, highlighting its potential as a therapeutic target for mitigating age-related effects.

## 1. Introduction

Aging represents a multifaceted biological process characterized by cascades of molecular changes within an organism, ultimately resulting in a progressive decline in physiological integrity and function. This decline manifests as a heightened susceptibility to mortality and a spectrum of age-related diseases, encompassing specific cancers [1,2] and cardiovascular issues [3], but also neurodegenerative conditions like macular degeneration [4], Alzheimer’s disease [5], or osteoporosis [6]. As the proportion of older individuals within human populations continues to rise, understanding the mechanisms driving age-related morbidity becomes increasingly critical from a clinical standpoint. Recent evidence underscores the impact of aging on key cellular processes. Firstly, mechanisms involved in DNA repair become less efficient with age, leading to an accumulation of damage exacerbated by environmental stresses [7,8,9]. Secondly, alterations in transcriptional profiles occur, with genes silenced in youth becoming activated in aged organisms [10,11]. Additionally, mechanisms governing genome stability exhibit decreased efficiency, resulting in genomic instability at heterochromatic loci [11,12]. Despite significant advancements in the field, the precise regulatory mechanisms of aging remain incompletely understood.

*Drosophila melanogaster* emerges as an invaluable model organism for investigating aging due to its short lifespan [13]. Our previous work unveiled the critical role of the poly(ADP–ribose) glycohydrolase PARG and poly(ADP–ribose) polymerase PARP1 in transcriptional regulation during this stage [14,15]. These enzymes operate within the poly(ADP–ribose) pathway, where PARP1 synthesizes poly(ADP–ribose) chains at the surface of chromatin proteins, resulting in their dissociation from DNA and subsequent chromatin opening, while PARG reverses PARP1 activity by removing poly(ADP–ribose) moieties from acceptor proteins [16,17,18,19,20,21]. We found that PARG, in conjunction with PARP1, facilitates the expression of developmental genes while repressing the expression of metabolic genes independently of PARP1 during the end of the third instar larval stage [14,15]. Collectively, these discoveries suggest that PARP1 and PARG may play a regulatory role in the expression of aging-related genes during later developmental stages. However, how PARG activity is regulated is poorly understood both in mammals and in *Drosophila*.

In this study, we explore the role of PARG in the transcriptional regulation of the aging process in *Drosophila melanogaster*. We report that the uncharacterized PARG C-terminal domain is a regulatory domain that controls PARG activity. The absence of this C-terminal regulatory domain is not lethal but profoundly affects the lifespan of animals, correlating with a misregulation of age-related genes that occurs earlier than in wild-type animals during the aging process, suggesting a premature aging of animals lacking the PARG C-terminal regulatory domain. Overall, our results show that PARG modulates the expression of age-related genes to control longevity.

## 2. Results

### 2.1. PARG Controls the Expression of Aging Genes during Third Instar Larval Stage

Building upon previous studies, we identified a comprehensive set of 1184 genes displaying significant dysregulation associated with aging in *Drosophila* [22,23,24]. Our investigation illuminated three primary functional shifts during aging progression, as depicted in Figure 1: a decrease in genes encoding muscle structural components (Figure 1A) and an increase in genes involved in cell death (Figure 1B) and defense responses (Figure 1C), underscoring the profound impact on transcriptional profiles during aging progression (Figure 1D) [24].

Interestingly, the expression of age-related genes extends beyond adulthood to other developmental stages, particularly at the end of the third instar larval stage. Intriguingly, our investigations revealed that in the absence of PARG, age-related genes exhibit heightened dysregulation at the end of the third instar larval phase (Figure 1E) [15]. Specifically, among the 27 age-related genes encoding muscle structural components, 21 demonstrated significant misregulation in *parg* mutants at this stage (Appendix A). Similarly, 4 out of 10 age-related genes involved in cell death processes exhibited significant misregulation in *parg* mutants (Appendix A). Additionally, 21 out of 27 age-related genes associated with toxin resistance, a subgroup of defense response genes, displayed significant misregulation in *parg* mutants (Appendix A).

Furthermore, our observations revealed that PARG forms binding associations with 734 age-related genes, encompassing 62.2% of this gene subset, during the end of the third instar larval phase [14]. Notably, among these targets, we identified pivotal regulators of *Drosophila* cell death pathways, Damm and Decay (Figure 1F) [25]. Remarkably, Damm and Decay exhibited substantial upregulation both during aging and in *parg* mutant larvae at the end of the third instar stage [15]. Additionally, our investigation unveiled PARG’s binding affinity to genomic regions associated with muscle structural components in *Drosophila*, known for their downregulation during aging [23,24]. Specifically, we identified PARG binding to Act88F and TpnC47D loci (Figure 1F), both of which display dysregulated expressions in *parg* mutant larvae (−1.91 and 14.18, respectively). Collectively, these findings illuminate the pivotal role of PARG in modulating the expression profile of age-related genes during the third instar larval phase, hinting at its potential involvement in governing these processes during adult aging.

### 2.2. Characterization of Drosophila PARG C-Terminal Domain

Next, we investigated the structure of the PARG protein in mammals and in *Drosophila*. Human PARG (hPARG) is a 976-amino acid protein characterized by a regulatory domain (1–360) and a catalytic domain (361–976), including the PARG signature (706–825) [26,27] (Figure 2A). In contrast, the *Drosophila melanogaster* PARG protein (dPARG) is a 768-amino acid protein consisting of an N-terminal domain with catalytic activity (1–606) (hereafter referred to as the catalytic domain), encompassing the PARG signature (334–455) that harbors essential residues for catalytic activity (E385, E386, and E394) [26] (Figure 2A). Notably, this domain is well conserved among mammalian and *Drosophila* species [26] (Figure 2A). While the regulatory domain of hPARG is absent in dPARG, intriguingly, dPARG possesses a C-terminal domain (607–768) absent in hPARG.

### 2.3. PARG C-Terminal Domain Is Critical for PARG Catalytic Activity

To elucidate if the *Drosophila* PARG C-terminal domain possesses a regulatory function, we generated a GFP–tagged version of the full-length PARG protein (referred to hereafter as PARG^WT^) and a GFP–tagged variant lacking its C-terminal domain (606–768 region) (referred to hereafter as PARG^∆Cter^).

(Figure 2B) Expression of both PARG^WT^ and PARG^∆Cter^ in a *parg* null mutant background successfully rescued the developmental arrest observed at the end of third instar larval stage in *parg* mutant animals.

PARP1 is highly active during late third instar larvae leading to a consequential pADPr accumulation if functional PARG is absent [20]. To elucidate the contribution of the C-terminal domain to PARG catalytic activity, we evaluated pADPr levels across the PARG^WT^ and PARG^∆Cter^ variants. Remarkably, the PARG^∆Cter^ variant exhibited a significant increase in pADPr levels compared to PARG^WT^ when endogenous PARG was lacking (Figure 2C,D) (Appendix A), implying the indispensability of the C-terminal domain for PARG catalytic activity. These findings collectively affirm the critical role of the PARG C-terminal domain in catalytic activity.

We previously reported that the disruption of PARG phosphorylation sites contributes to a decrease in the stability of the PARG protein [26]. To investigate whether the C-terminal domain influences PARG protein stability, we conducted a Western blot analysis. PARG^WT^ exhibited a single band at 108 kDa (Figure 2C). Intriguingly, PARG^∆Cter^ presented two bands migrating around the expected size of 91 kDa, with the higher band being more pronounced (Figure 2C). Importantly, PARG^∆Cter^ exhibited comparable levels of PARG protein relative to PARG^WT^ (Figure 2E). These findings collectively affirm the critical role of the PARG C-terminal domain in catalytic activity but does not influence PARG protein stability. Overall, we identified the PARG C-terminal domain as the PARG regulatory domain.

### 2.4. The Absence of PARG C-Terminal Regulatory Domain Affects Adult Longevity

Given PARG’s regulatory role in the expression of age-related genes during the late third instar larvae stage, it is plausible that PARG influences adult longevity. In line with human trends, wild-type *D. melanogaster* typically exhibits a convex survivorship curve [13], characterized by low mortality rates in early and middle life, followed by a sharp increase in mortality with age. Females expressing PARG^WT^ display a survivorship curve consistent with this pattern (Figure 3A). However, females expressing the PARG^∆Cter^ variant exhibit a distinct survivorship curve shape, marked by higher mortality rates even in the early days of adulthood (Figure 3A). Furthermore, the survivability rate of females expressing PARG^∆Cter^ is notably affected. While fifty percent of PARG^WT^–expressing females are alive at Day 42, PARG^∆Cter^–expressing females reach 50% of survivability at Day 25.

Male populations demonstrate a similar phenotype, with PARG^WT^–expressing males exhibiting similar convex survivorship curves, while PARG^∆Cter^–expressing males display an almost linear survivorship curve, with fifty percent of the male population alive at Day 22 compared to Day 31 for PARG^WT^–expressing males (Figure 3B). Additionally, we observed that the PARG^WT^ and PARG^∆Cter^ variants exhibit similar survivorship curves in a wild-type background (Appendix A), suggesting that the effect of the longevity is due to the absence of the PARG C-terminal regulatory domain rather than an overexpression of the PARG catalytic domain. Taken together, these findings underscore the detrimental effect of the absence of the PARG C-terminal regulatory domain on longevity.

### 2.5. PARG Governs Temporal Gene Expression Dynamic in Aging

Next, we investigated the impact of the absence of the PARG C-terminal regulatory domain on the regulation of age-related genes during the aging process. Notably, we observed a pronounced effect on longevity in females compared to males (Figure 3A,B), prompting our focus on the female population. We designated Day-5 females as representative of the young adult cohort and examined timepoints at Day 15 and Day 25. This decision stemmed from the similarity in survival rates between PARG^∆Cter^ females at Day 15 and PARG^WT^ females at Day 25 (Figure 3A,B).

Our investigation revealed that the two crucial components of cell death pathways, Damm and Decay, exhibited a progressive increase in expression levels throughout aging in PARG^WT^ animals (Figure 3C,D). Conversely, animals expressing the PARG^∆Cter^ variant exhibited elevated expression of the Damm gene from Day 5 to Day 25 (Figure 3C). Particularly noteworthy was the similarity in Damm expression levels between Day-5 PARG^∆Cter^ and Day-25 PARG^WT^ animals. While Decay expression at Day 5 showed no significant difference between the PARG^∆Cter^ and PARG^WT^ animals (Figure 3D), a substantial increase was observed in Day-15 PARG^∆Cter^ animals, akin to the levels seen in Day-25 PARG^WT^ animals, while Day-15 PARG^WT^ animals did not show a similar increase.

Further investigation revealed that the expression of the *Drosophila* defense response gene Cyp4p3 remained relatively stable between Day 5 and Day 15 in PARG^WT^ animals but increased threefold by Day 25 (Figure 3E). Conversely, Day-5 PARG^∆Cter^ animals exhibited a twofold increase in Cyp4p3 expression compared to Day-5 PARG^WT^ animals, with no significant change in expression observed thereafter. This suggests a loss of temporal regulation in Cyp4p3 expression in the absence of the PARG C-terminal domain.

Lastly, we examined the expression of muscle structure components Act88F, TpnC47D, and Obsc, noting a progressive decline over time in PARG^WT^ animals (Figure 3F–H). Although Day-5 PARG^∆Cter^ animals showed a slight, nonsignificant decrease compared to Day-5 PARG^WT^ animals in the expression of Act88F, TpnC47D, and Obsc, Day-15 PARG^∆Cter^ animals exhibited a significant decrease in Act88F and TpnC47D expression compared to their PARG^WT^ counterparts (Figure 3F,G). Additionally, Day-15 PARG^∆Cter^ animals displayed decreased Obsc expression compared to PARG^WT^, although the difference was not statistically significant. By Day 25, the expression levels of Act88F, Tpnc47D, and Obsc were similar between the PARG^∆Cter^ and PARG^WT^ animals.

Taken together, our findings underscore the essential role of PARG in temporally regulating the expression of age-related genes during the aging process.

## 3. Discussion

In this manuscript, we unveiled the regulatory role of the previously uncharacterized PARG C-terminal domain in governing PARG activity. Additionally, we established a significant association between the poly(ADP–ribose) pathway and the aging process, with PARG regulating the expression of age-related genes both during the end of the third instar larval and during adulthood. Our previous findings indicated that PARG^WT^-GFP is expressed at levels 1.6-fold higher than endogenous PARG [14]. Interestingly, flies expressing PARG^WT^-GFP in a *parg* mutant background or those expressing PARG^ΔCter^-GFP in a wild-type background exhibit similar lifespans to flies expressing PARG^WT^-GFP in a wild-type background. This suggests that the reduced lifespan observed in flies expressing PARG^ΔCter^-GFP is due to the absence of the PARG regulatory domain rather than an overexpression of the PARG catalytic domain. 

Additionally, we found that PARG controls the expression of age-related genes involved in the cell death process, a process tightly related to aging both in *Drosophila* and mammals [28,29]. Specifically, we found that PARG directly binds to the Damm and Decay loci—two caspases that are the final effectors of cell death [25]—and that the absence of the PARG C-terminal domain leads to the premature overexpression of Damm and Decay suggests a premature activation of the cell death process and involvement of PARG in the regulation of cell death processes during aging.

Our prior research has shown that the activity of the poly(ADP–ribose) pathway fluctuates significantly across developmental stages [20]. Additionally, we observed a progressive increase in poly(ADP–ribose) levels during aging, with Day-50 adults displaying a 1.4-fold increase compared to Day-3 animals (Appendix A). Collectively, our findings support a model wherein, in young wild-type adults, PARG facilitates the expression of genes coding for muscle structure components while suppressing the expression of genes involved in cell death and defense responses (Figure 4A). As aging ensues, PARG activity diminishes gradually, leading to the progressive misregulation of age-related genes, consistent with the escalating poly(ADP–ribose) levels during adult aging (Appendix A). However, disruption of PARG accelerates the dysregulation of gene expression, resulting in a premature aging phenotype that detrimentally affects lifespan. 

The aging process responds to stress cues that can culminate in the accumulation of structural damages [7,8,9], yet it also relies on molecular clocks such as DNA methylation levels, which serve as reliable indicators of an individual’s age [30] (Figure 5). Moreover, while aging was conventionally perceived as the mere accumulation of molecular damages, recent studies suggest that aging might also follow predetermined patterns governed by developmental processes [31]. Our discovery prompts us to hypothesize that the poly(ADP–ribose) pathway may act as an intermediary by modulating the expression of age-related genes under the influence of various cues. This hypothesis finds support in recent findings: Firstly, PARP1 is crucial for the activation of heat shock genes during stress, indicating that stress cues can modulate PARP1 activity [18,32,33]. Secondly, recent studies have unveiled PARP1’s role in the circadian clock, with its activity oscillating in the mammalian liver on a daily rhythm controlled by feeding patterns [34]. Lastly, we have recently demonstrated that PARP1 and PARG regulate the expression of developmental and metabolic genes during the latter stages of the third instar larval stage [14,15]. Interestingly, we observed that PARG functions as a transcriptional activator only in the presence of PARP1; in its absence, PARG is solely involved in transcriptional repression. This suggests that PARP1 may be implicated in promoting the expression of genes coding for muscle structure components in young adult animals. Additionally, we note that the presence of PARG represses the expression of genes coding for muscle structure components during the latter stages of the third instar larval stage while promoting their expression during adulthood, indicating a degree of flexibility in PARG’s function.

Additionally, the PARG regulatory domain we identified in *Drosophila* is absent in mammals. In mammals, an N-terminal domain, which is absent in *Drosophila*, plays the role of a regulatory domain [35,36]. However, the structure of the *Drosophila* PARG regulatory domain differs from that of the mammalian PARG regulatory domains [26]. Therefore, we cannot exclude the possibility that the regulation of PARG activity during aging in mammals differs from the mechanism we uncovered in this manuscript.

Furthermore, our findings reveal a more pronounced reduction in lifespan for female flies compared to male flies in the absence of the PARG regulatory domain. This compelling observation suggests potential sex-specific differences in the regulation of the PARG protein. Although we did not explore these differences in the current manuscript due to time and scope constraints, they represent a promising avenue for future research. Investigating these sex-specific regulatory mechanisms could unveil critical insights and potentially transformative discoveries in the field of aging and molecular biology.

Taken together, our results underscore the critical role of the poly(ADP–ribose) pathway in the aging process and highlight its potential as a target for future treatments aimed at mitigating the effects of aging.

## 4. Materials and Methods

### 4.1. Drosophila Strains and Genetics

Flies were grown at 20 °C unless otherwise stated. The transgenic stock with P{w1, UASt::PARG-EGFP}, called PARG^WT^ in this study, was described in [20]. The *parg^27.1^* mutant was described in [19] and *69B-GAL4* driver was described in [37].

### 4.2. Generation of Drosophila PARG Variants

PARG^∆Cter^ was generated by PCR from genomic DNA using the following primers:

Forward primer: AAGGTACCATGCAAGAATTCAGGTCACACTTG

Reverse primer: TTGGATCCGGCGGATGCTCCCTC

Amplicon includes 5′UTR and intronic region and contains the 1–606 region. The insert was cloned into pUASt-EGFP vector.

### 4.3. Western Blot

The following antibodies were used for immunoblotting assays: pADPr reagent (MABE1031; Rabbit 1:2000; Sigma-Aldrich, St Louis, MO, USA), anti-GFP (Jl-8 Mouse monoclonal, #632380, 1:4000; BD Biosciences, San Diego, CA, USA), anti-H4 (H-97; Rabbit polyclonal; 1:1000; sc-10810; Santa Cruz, Dallas, TX, USA), and anti-H3 (FL-136; Rabbit polyclonal; 1:1000; sc-10809; Santa Cruz, Dallas, TX, USA). Western blotting was performed using the detection kit from GE Healthcare, Amersham, UK (#RPN2106), according to the manufacturer’s instructions.

### 4.4. Adult Lifespan Measurement

A total of 20 virgin females and 20 virgin males for each condition were place in a tube containing standard cornmeal–molasses–agar media with dry yeast. The experiment was performed in triplicate. The flies were transferred to a fresh tube every two days and the deaths were counted daily. The flies were grown at 20 °C.

### 4.5. Quantitative RT-PCR Assays

This assay was performed in duplicate. A total of 20 virgin males and 20 virgin females were gathered for each condition. Three female flies, expressing either PARG^WT^ or PARG^∆Cter^ in a *parg* mutant background were collected at three different time points: D5, D15, and D25. Total RNA was purified using the RNeasy kit (Qiagen, Hilden, Germany). cDNA was obtained by reverse transcription using M-NLV reverse transcriptase (Invitrogen, Carlsbad, CA, USA) using 300 ng of purified RNA for each sample. The obtained cDNAs were diluted ten times in distilled water. Real-time PCR assays were performed on an Applied Biosystems StepOnePlusTM instrument and using Luna^®^ Universal qPCR Master Mix (New England Biolabs, Ipswich, MA, USA). A total of 4 µL of cDNA was used for each sample to a final volume of 10µL. The final concentration of each primer was 0.5 µM. The amount of DNA was normalized using the difference in threshold cycle (CT) values (ΔCT) between rpL32 and target genes.

The quantitative real-time PCR (qPCR) primer sequences for *D. melanogaster* ribosomal protein L32 gene (RpL32) were 5′-GCTAAGCTGTCGCAACAAAT-3′ (forward) and 5′-GAACTTCTTGAATCCGGTGGG-3′ (reverse).

Sequences for Decay were 5′-CTTCGACGATCTGACCTTCTC-3′ (forward) and 5′-CATCACCGCCAACACAAAG-3′ (reverse).

Sequences for Damm were 5′-TGTATCTGCCCGAAAGAACAG-3′ (forward) and 5′-CTGAGGGAACTGCTCATGATT-3′ (reverse).

Sequences for cyp4p3 were 5′-GCTTCGATTTGTACGGCAAAG-3′ (forward) and 5′-CGCACTATCTGCGTCTATAACA-3′ (reverse).

Sequences for Ninad were 5′-GCCCCACATTTACCTTCATTG-3′ (forward) and 5′-AGAGATGTCCACCATTCGC-3′ (reverse).

Sequences for Act88f were 5′-AATCGAACGTGCGACTCTATC-3′ (forward) and 5′-ACTAATGCACCCGCATCAT-3′ (reverse).

Sequences for TpnC47D were 5′-CTCGACATCATGATTGAGGAAATC-3′ (forward) and 5′-TGGGTCCACAATGCTTACTC-3′ (reverse).

Sequences for Obsc were 5′-GATGAAGGCCACCAACTTTATTG-3′ (forward) and 5′-GTGGCGATGACTTCGTACTT-3′ (reverse).

### 4.6. Statistical Analysis

The significance of non-categorical data presented in Figure 1A–C were addressed with a paired two-tailed *t*-test; differences were considered strongly significant when the *p*-value was lower than 0.01.

The significance of non-categorical data presented in Figure 1E was addressed with an unpaired two-tailed *t*-test; differences were considered strongly significant when the *p*-value was lower than 0.01.

Quantification of the Western blot presented in Figure 2C-E was performed using Fiji based on three independent blots. The significance was addressed by an unpaired two-tailed *t*-test; differences were considered strongly significant when the *p*-value was lower than 0.01 and non-significant when *p*-value was higher than 0.05.

The significance of non-categorical data presented in Figure 3 was addressed with an unpaired two-tailed *t*-test; differences were considered strongly significant when the *p*-value was lower than 0.01, significant when *p*-value was between 0.05 and 0.01, and non-significant when *p*-value was higher than 0.05.

## Figures and Tables

**Figure 1 ijms-25-06189-f001:**
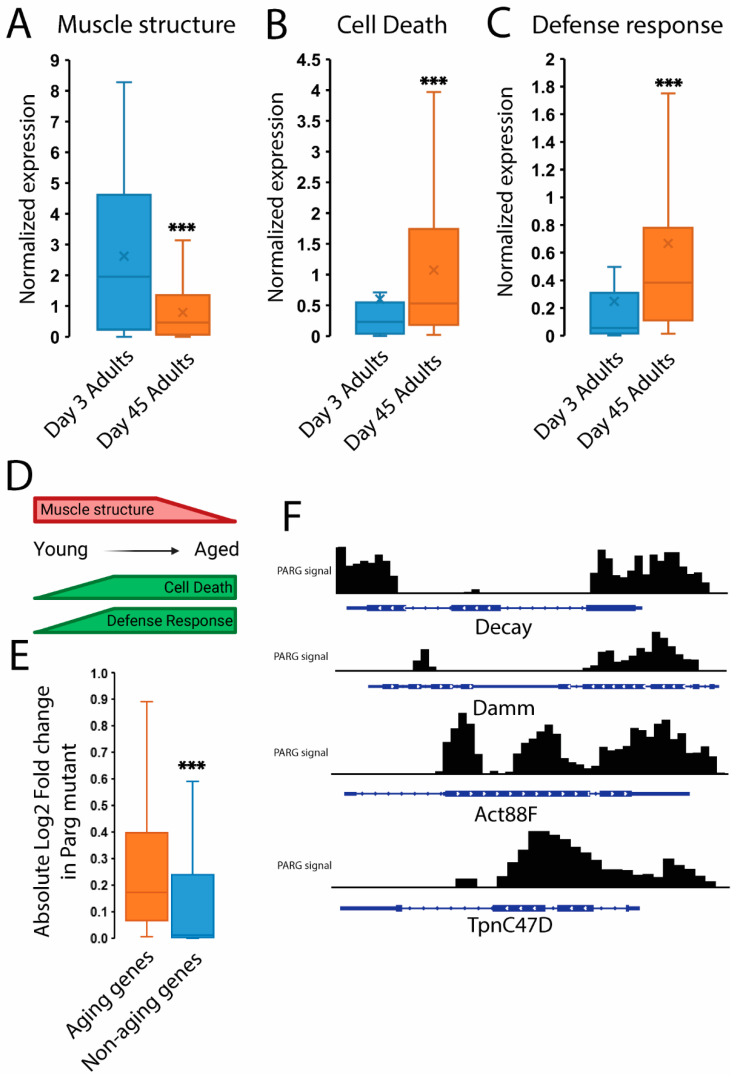
Expression of age-related genes during the aging process and at the third instar larval stage. (**A**–**C**) Box plots illustrating the expression levels of genes associated with muscle structure (**A**), cell death processes (**B**), and defense responses (**C**). Expression levels are compared between Day 3 and Day 45 adult flies, represented in blue and orange, respectively. Data are sourced from our previous investigation on aging-related transcriptional changes in *D. melanogastger* [24], with gene expression normalized to the average of Day 5 adult flies. Statistical analysis was conducted via paired two-tailed *t*-tests, denoted by *** for *p*-values < 0.01. The list of the genes used for these graphs can be found in Appendix A. (**D**) Schematic depicting the major processes affected during animal aging, highlighting the significant downregulation of muscle structure genes and upregulation of cell death and defense response genes. (**E**) Box plot displaying the absolute fold change, depicted on a log2 scale, observed between *parg* mutant and wild type animals at the end of the third instar larval stage. Fold changes of age-related genes are shown in blue, while those unrelated to aging are shown in orange. Data are sourced from [15], with statistical analysis performed using unpaired two-tailed *t*-tests and denoted by *** for *p*-values < 0.01. (**F**) Integrated Genome Viewer (IGV) track illustrating the distribution of the PARG protein along key loci involved in cell death processes (Decay and Damm) and muscle structure (Act88F and TpnC47D). Data are sourced from [14].

**Figure 2 ijms-25-06189-f002:**
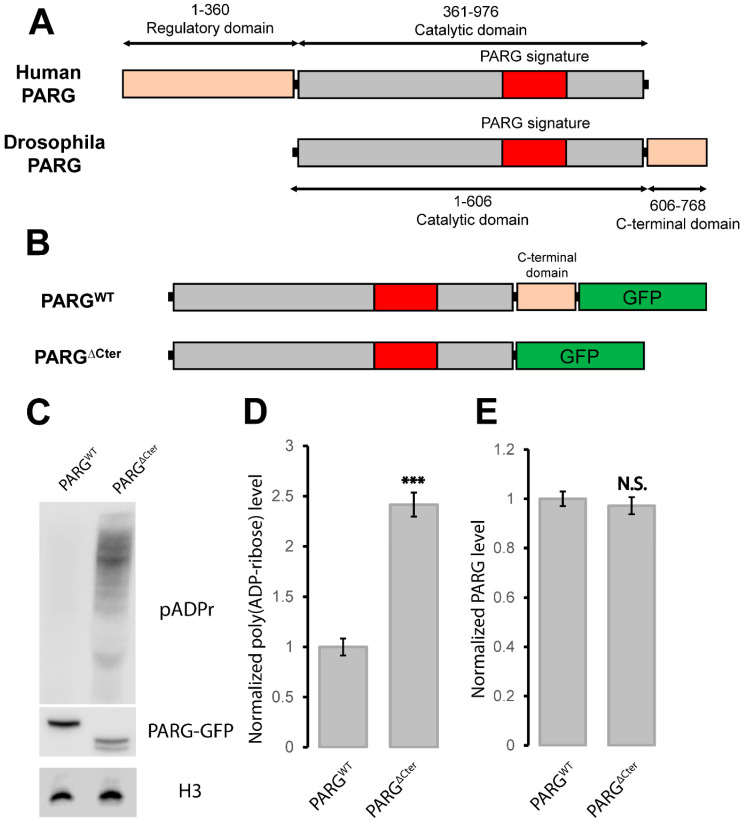
PARG C-terminal domain controls PARG activity. (**A**) Schematic illustration comparing Human and *Drosophila* PARG proteins. Human PARG features a distinctive 360-amino acid regulatory domain absent in *Drosophila* species, while the catalytic domain is evolutionarily conserved. The PARG signature, essential for catalytic activity, is highly conserved across human and *Drosophila* species [26]. *Drosophila* PARG exhibits a C-terminal region of 162 amino acids absent in human PARG. (**B**) Overview of engineered PARG variants generated in this study. PARG^WT^ represents a GFP–tagged full-length PARG and PARG^∆Cter^ is a GFP–tagged PARG lacking the C-terminal domain (606–768 region). (**C**) Western blot analysis depicting the poly(ADP–ribose) (pADPr) levels (**top** panel), GFP expression (**middle** panel), and Histone H3 as a loading control (**lower** panel) for different PARG variants. PARG^WT^ and PARG^∆Cter^, are expressed in a *parg* null mutant background. (**D**,**E**) Quantitative assessment of pADPr levels (**D**) and PARG protein abundance based on GFP level (**E**) based on three independent blots. Statistical analysis was conducted using an unpaired two-tailed *t*-test. N.S indicates non-significant results while *** indicates a *p*-value < 0.01.

**Figure 3 ijms-25-06189-f003:**
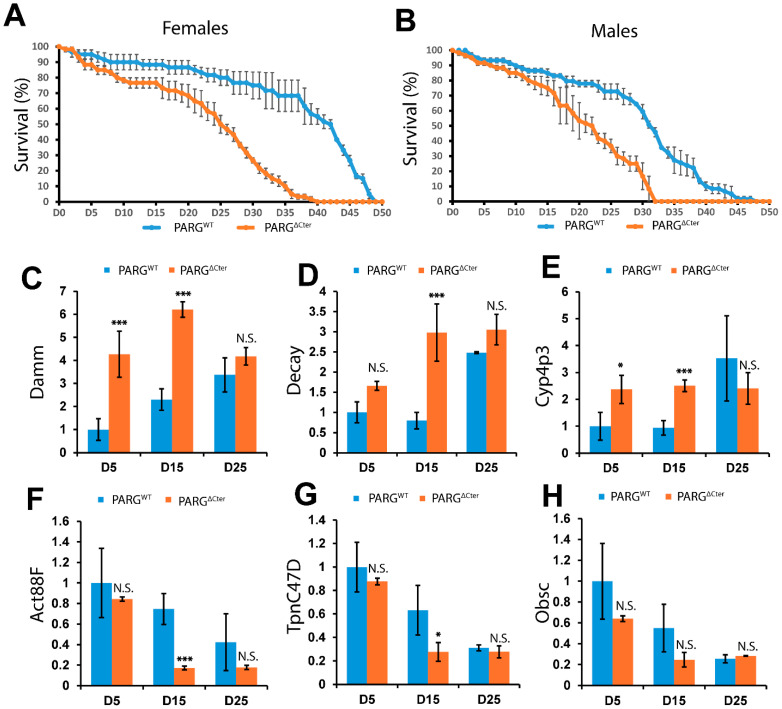
The absence of PARG C-terminal domains affects the expression of age-related genes and lifespan. (**A**,**B**) Lifespan curves depict female (**A**) and male (**B**) adult populations expressing either PARG^WT^ (blue) or PARG^∆Cter^ (orange) in a *parg* mutant background. Y-axis represents the percentage of flies surviving on specific days post-hatching, with Day 0 denoting adult emergence. Triplicate experiments were conducted, and error bars represent standard error of mean (SEM). (**C**–**H**) Expression levels of key components involved in cell death processes (Damm and Decay), defense response (Cyp4p3), and muscle structure (Act88F, TpnC47D, and Obsc) at Day 5, Day 15, and Day 25 in adult females. Flies expressing PARG^WT^ (Blue) or PARG^∆Cter^ (Orange) in a *parg* mutant background were analyzed. For each panel, the expression is normalized to the expression of Day-5 PARG^WT^. Duplicate experiments were performed, with error bars indicating SEM. Statistical analysis utilized unpaired two-tailed *t*-tests, with significance denoted by ***** for *p*-value < 0.05, *** for *p*-value < 0.01, and N.S. for non-significant results.

**Figure 4 ijms-25-06189-f004:**
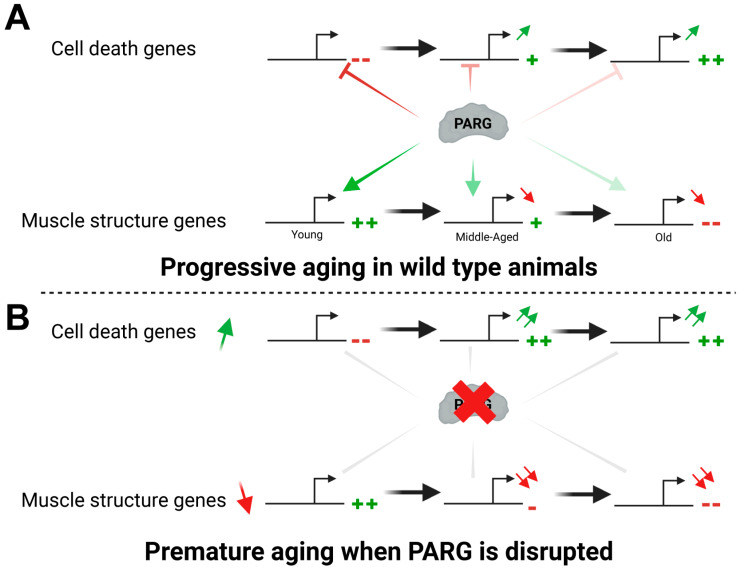
PARG modulates the expression of age-related genes and lifespan. Schematic illustrating the regulation of age-related gene expression during the aging process in animals. (**A**) Initially, upon reaching adulthood, PARG presence promotes the expression of genes related to muscle structure while repressing genes associated with cell death and defense response processes. However, as animals age, PARG activity is progressively suppressed, resulting in a gradual downregulation of muscle structure genes and concurrent upregulation of genes linked to cell death and defense responses. (**B**) In the absence of functional PARG, this progressive regulation of age-related genes is disrupted, leading to an accelerated misregulation of these genes. Consequently, animals lacking functional PARG experience premature aging, culminating in a shortened lifespan. --: Gene expression is very low. -: Gene expression is low. +: Gene expression is high. ++: Gene expression is very high.

**Figure 5 ijms-25-06189-f005:**
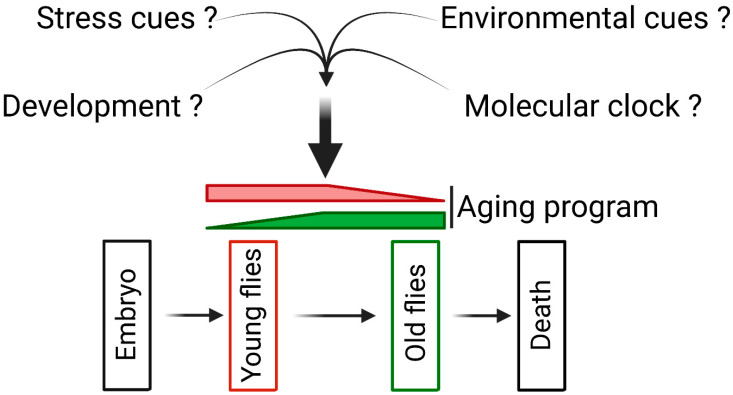
The multifaceted control of the aging program. This schematic depicts the intricate regulation of age-related gene expression by diverse external factors, encompassing stressors, environmental cues, developmental processes, and molecular clock mechanisms. Our findings propose that the poly(ADP–ribose) pathway potentially acts as a crucial intermediary, finely tuning the aging program in response to the influence of these varied cues. The question marks indicate that the specific external factors affecting the aging program remain unclear.

## Data Availability

Mutant strains and transgenic stocks are available upon request. The authors state that all data necessary to confirm the conclusions presented in the article are represented fully within the article. The PARG ChIP-seq raw and processed data are accessible upon demand or on the GEO platform: GSE228898. The PARG RNA-seq raw and processed data are accessible upon demand or on the GEO platform: GSE200499.

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
