# Peer review of "PARG Protein Regulation Roles in Drosophila Longevity Control"

_ijms, 2024, doi:10.3390/ijms25116189_

Round 1
Reviewer 1 Report
Comments and Suggestions for Authors
The manuscript titled “PARG Protein Regulation Roles in Longevity Control” assesses the role of PARG in the aging process, primarily regarding gene expression and longevity, using the experimental model Drosophila melanogaster. It covers an intriguing topic, but I recommend major revisions before considering the publication of this manuscript in IJMS. Overall, the manuscript is well-written with interesting data, but I find its structure and discussion lacking.
- Authors have included results description and figures in the Introduction section. That only should be in the Results Section. (I don’t know if I’m right, but I understand that Figure 1 contains previously published results). At the end of the introduction section, authors should briefly mention the main aim of the work and highlight the main conclusions.
- Similarly, in the discussion section, references to figures or tables are generally avoided (except when results and discussion are combined in one section). The paragraphs between lines 223-244 represent a summary of the results section. In my opinion, this information should be integrated with the rest of the discussion and compared with more bibliographic references. In other words, the discussion should be enhanced. For instance, specific genes related to aging may be mentioned and discussed.
Other minor comments:
- In the Figure 1. Which one are the genes associated with muscle structure (A), cell death processes (B), and defense responses (C)?
- Drosophila melanogaster should be written in italics (for instance, in lines 67 or 156). Furthermore, it could be abbreviated to D. melanogaster second and successive times.
- According to guidelines for authors, “title should identify if the study reports (human or animal) trial data, or is a systematic review, meta-analysis or replication study.” Therefore, authors should modify the title and add that information.
- Maybe authors could include Drosophila melanogaster in the keywords to enhance its visibility in searches.
Author Response
Reviewer 1:
The manuscript titled “PARG Protein Regulation Roles in Longevity Control” assesses the role of PARG in the aging process, primarily regarding gene expression and longevity, using the experimental model Drosophila melanogaster. It covers an intriguing topic, but I recommend major revisions before considering the publication of this manuscript in IJMS. Overall, the manuscript is well-written with interesting data, but I find its structure and discussion lacking.
R1: Authors have included results description and figures in the Introduction section. That only should be in the Results Section. (I don’t know if I’m right, but I understand that Figure 1 contains previously published results). At the end of the introduction section, authors should briefly mention the main aim of the work and highlight the main conclusions.
Response: We deeply appreciate Reviewer 1's careful reading of our work and their invaluable comments, which have significantly contributed to the improvement of our manuscript. We have extensively revised the Introduction and Results sections to address the reviewer's points. All figure descriptions have now been relocated to the Results section. Additionally, we have modified the end of the Introduction to clearly state the aims of our study and highlight the main conclusions.
R1: Similarly, in the discussion section, references to figures or tables are generally avoided (except when results and discussion are combined in one section). The paragraphs between lines 223-244 represent a summary of the results section. In my opinion, this information should be integrated with the rest of the discussion and compared with more bibliographic references. In other words, the discussion should be enhanced. For instance, specific genes related to aging may be mentioned and discussed.
Response: We thank Reviewer 1 for this valuable suggestion. We have extensively revised the Discussion section to improve its flow and coherence. The previously summarized results between lines 223-244 have been integrated with the rest of the discussion. Furthermore, we have expanded our discussion on the involvement of PARG in regulating cell death processes in Drosophila. We believe these modifications will comprehensively address the reviewer's concerns and enhance the overall quality of the manuscript.
R1: In the Figure 1. Which one are the genes associated with muscle structure (A), cell death processes (B), and defense responses (C)?
Response: We thank Reviewer 1 for their question. The lists of age-related genes involved in muscle structure, cell death processes, and defense responses are provided in Supplemental Figures 1, 2, and 3, respectively. We have also clarified this in the figure legend for Figure 1 to ensure that it is easy to locate the relevant information.
R1: Drosophila melanogaster should be written in italics (for instance, in lines 67 or 156). Furthermore, it could be abbreviated to D. melanogaster second and successive times.
Response: We have revised the manuscript to italicize Drosophila melanogaster where necessary and abbreviated it to D. melanogaster after its first mention.
R1: According to guidelines for authors, “title should identify if the study reports (human or animal) trial data, or is a systematic review, meta-analysis or replication study.” Therefore, authors should modify the title and add that information.
Response: We modified the title: “PARG Protein Regulation Roles in Drosophila Longevity Control” to better fit our study's focus and adhere to the guidelines.
R1: Maybe authors could include Drosophila melanogaster in the keywords to enhance its visibility in searches.
Response: We have included Drosophila melanogaster in the keywords. We thank Reviewer 1 again for their careful reading and valuable suggestions to enhance our work.
Reviewer 2 Report
Comments and Suggestions for Authors
The article titled "PARG Protein Regulation Roles in Longevity Control" seeks to elucidate the intricate molecular mechanisms governing the pADPr pathway in the aging process.
The authors provided a comprehensive analysis of the role of PARG in regulating the expression of age-related genes during the aging process in Drosophila.
The authors provided new evidence on the relevance of these molecular mechanisms in the aging process, opening new horizons in understanding various diseases, ranging from age-related pathologies to cancer therapy, thereby improving therapeutic approaches.
The references cited were aptly selected, reflecting contemporary literature pertinent to the research domain.
However, several points could be improved:
- The study exclusively focuses on Drosophila. Therefore, I suggest modifying the manuscript title to better reflect the conducted research.
- A section on the statistical analysis used is completely missing. The authors should include such a section in the materials and methods.
- References in the text regarding the expression of aging-related genes should be contextualized to the animal model used, even if these genes are highly conserved between Drosophila and mammals, as there exists a complex and intricate gap between the genes involved in this delicate physiological process in the two species.
- The authors should incorporate the study's limitations into the discussion.
Author Response
Reviewer 2:
The article titled "PARG Protein Regulation Roles in Longevity Control" seeks to elucidate the intricate molecular mechanisms governing the pADPr pathway in the aging process.
The authors provided a comprehensive analysis of the role of PARG in regulating the expression of age-related genes during the aging process in Drosophila.
The authors provided new evidence on the relevance of these molecular mechanisms in the aging process, opening new horizons in understanding various diseases, ranging from age-related pathologies to cancer therapy, thereby improving therapeutic approaches.
The references cited were aptly selected, reflecting contemporary literature pertinent to the research domain.
However, several points could be improved:
R2: The study exclusively focuses on Drosophila. Therefore, I suggest modifying the manuscript title to better reflect the conducted research.
Response: We thank Reviewer 2 for their invaluable comments, which have significantly contributed to the improvement of our manuscript. We have modified the title to: “PARG Protein Regulation Roles in Drosophila Longevity Control” to better reflect the focus of our research.
R2: A section on the statistical analysis used is completely missing. The authors should include such a section in the materials and methods.
Response: We thank Reviewer 2 for pointing out this omission. We have revised the manuscript to include a section on the statistical analysis used in the Materials and Methods section.
R2: References in the text regarding the expression of aging-related genes should be contextualized to the animal model used, even if these genes are highly conserved between Drosophila and mammals, as there exists a complex and intricate gap between the genes involved in this delicate physiological process in the two species.
Response: We thank Reviewer 2 for noticing this omission. We have extensively revised the manuscript, particularly the Introduction, Results, and Discussion sections, to provide better context regarding the animal model we referred to in each specific section. We hope these revisions adequately address Reviewer 2’s concerns. Thank you once again for your valuable feedback.
R2: The authors should incorporate the study's limitations into the discussion.
Response: We thank Reviewer 2 for their insightful comment. We have extensively revised the Discussion section to incorporate a detailed discussion of the study's limitations. Specifically, we included a section discussing the differences between mammalian and Drosophila PARG. We believe these changes address Reviewer 2's concerns and provide a more comprehensive understanding of our study. Thank you once again for your careful reading and valuable feedback.
Reviewer 3 Report
Comments and Suggestions for Authors
The manuscript aimed to study the role of PARG in longevity control in Drosophila melanogaster. However, it mostly relies on reanalyzing data from published papers with minimal effort to generate new experimental data. This undermines its originality and contribution.
The functions of the PARG C-terminal domain are well-studied, given its importance, it is not surprising to see flies lacking this domain have reduced lifespan. To convincingly conclude that the PARG C-terminal domain controls longevity, the authors should demonstrate that its overexpression extends lifespan and corrects the dysregulated age-related genes. Much more work is needed in this regard.
There are many genes associated with cell death regulation in flies. The ten genes analyzed in this manuscript represent only a small fraction of them. Thus the conclusions on cell death dysregulation based on this small collection of genes are not comprehensive or compelling.
Also, I have to say the manuscript is not well written. The introduction, for example, focuses too much on the results but provides insufficient background information.
There is also a concern of overstatement. For example, the claim that “This research provides valuable insights into the underlying mechanisms of aging and opens avenues for the development of interventions to counteract its impact” is not supported by the limited data presented in tis manuscript.
For these reasons, I recommend that the manuscript be rejected.
Author Response
Reviewer 3:
The manuscript aimed to study the role of PARG in longevity control in Drosophila melanogaster. However, it mostly relies on reanalyzing data from published papers with minimal effort to generate new experimental data. This undermines its originality and contribution.
Response: We appreciate Reviewer 3's careful reading and comments on our manuscript. We would like to clarify that our study is entirely original. We are the first to report the role of the Drosophila PARG C-terminal domain as a regulatory domain of PARG activity and to identify the role of PARG in the transcriptional regulation of age-related genes. While our work builds upon and extends two of our previous studies (Bordet et al., Genes (Basel), 2021 and Bordet et al., Sci. Rep., 2023), it provides new experimental data and novel insights. We referenced these previous studies to provide context for our discoveries. We believe that our manuscript makes a significant and original contribution to the field by unveiling these new roles of PARG.
R3: The functions of the PARG C-terminal domain are well-studied, given its importance, it is not surprising to see flies lacking this domain have reduced lifespan. To convincingly conclude that the PARG C-terminal domain controls longevity, the authors should demonstrate that its overexpression extends lifespan and corrects the dysregulated age-related genes. Much more work is needed in this regard.
Response: We extend our gratitude to Reviewer 3 for their insightful comments. In the literature we only found one paper that briefly mentioned the Drosophila PARG C-terminal domain from Ivan Ahel’s laboratory (PMID: 34019811), without studying it in detail. Our study is the first to characterize the role of the Drosophila PARG C-terminal domain in regulating PARG activity.
Regarding the overexpression of PARG, the PARGWT-GFP version we use as a control line exhibits an expression level only 1.6-fold higher than endogenous PARG (Bordet et al., Sci. Rep., 2023). Furthermore, flies expressing PARGWT-GFP in an endogenous parg mutant or a wild-type background do not exhibit a significant difference in lifespan (Supplementary Fig. 4), suggesting that the overexpression of PARG does not affect lifespan. Additionally, the lifespans of flies expressing PARGΔCter-GFP or PARGWT-GFP in a wild-type background are similar, suggesting that the overexpression of PARG catalytic domain does not affect lifespan.
We have revised the discussion section to clarify these points. We believe that these changes address Reviewer 3’s concerns and provide a clearer understanding of our findings.
R3: There are many genes associated with cell death regulation in flies. The ten genes analyzed in this manuscript represent only a small fraction of them. Thus the conclusions on cell death dysregulation based on this small collection of genes are not comprehensive or compelling.
Response: We thank Reviewer 3 for highlighting this important point. We agree that the genes analyzed in this manuscript represent only a small fraction of the genes associated with cell death regulation in flies. Therefore, we cannot conclusively state that all age-related genes involved in cell death are directly regulated by PARG. While investigating the impact of PARG on the transcriptional regulation of age-related genes involved in cell death we carefully selected Damm and Decay that are two key caspases that act as the final effectors of cell death. Their premature overexpression in animals lacking the PARG C-terminal domain suggests an increased activity of cell death processes, indicating that PARG plays a critical role in regulating this process. We have revised the discussion section to clarify this point and to more accurately reflect the scope of our findings. We thank Reviewer 3 for their valuable feedback.
R3: Also, I have to say the manuscript is not well written. The introduction, for example, focuses too much on the results but provides insufficient background information.
Response: We have extensively revised the Introduction and Results sections to provide more comprehensive background information. All figure descriptions have been appropriately relocated to the Results section. We believe these modifications address Reviewer 3’s concerns and improve the clarity and quality of the manuscript.
R3: There is also a concern of overstatement. For example, the claim that “This research provides valuable insights into the underlying mechanisms of aging and opens avenues for the development of interventions to counteract its impact” is not supported by the limited data presented in tis manuscript.
For these reasons, I recommend that the manuscript be rejected.
Response: We thank Reviewer 3 for pointing out this overstatement and for their valuable contribution to improving our manuscript. We have revised the abstract section to remove the unsupported claim.
Round 2
Reviewer 2 Report
Comments and Suggestions for Authors
The authors have thoroughly addressed all the points previously highlighted in the manuscript. Each concern raised in the initial review has been revised, resulting in a significantly improved document.
Author Response
We want to express our gratitude for the invaluable comments from the reviewers and their constructive suggestions aimed at improving our manuscript. In response to their feedback, we have diligently revised and resubmitted our paper as an article, incorporating five primary figures, three supplementary tables, and two supplementary figures. It is essential to underline that this work represents a significant contribution to the field, presented here for the first time, with no prior publication.
Below we have provided a comprehensive response to the reviewer’s comments:
Reviewer 2:
The authors have thoroughly addressed all the points previously highlighted in the manuscript. Each concern raised in the initial review has been revised, resulting in a significantly improved document.
Response: We appreciate Reviewer 2 for their diligent review and constructive feedback, which has significantly contributed to enhancing the quality of our manuscript. Thank you for acknowledging the revisions made in response to the concerns raised during the initial review.
Reviewer 3 Report
Comments and Suggestions for Authors
The authors have made efforts to clarify the originality of their study, which builds on their previous work. The introduction and results sections have been improved, and unsupported claims have been removed. However, several issues and additional experiments are necessary to enhance the significance of the study and justify its publication.
1. There is a noticeable difference in the survival curves between males and females expressing PARGWT and PARGΔCter. While various possibilities could explain this, such as differentially regulated pathways in stress responses, immune function, and metabolic processes, one key question is: Do the expression levels of PARGWT and PARGΔCter vary between males and females? This could lead to the different survival outcomes. Can the authors provide data on the expression levels of PARGWT and PARGΔCter in both male and female Drosophila? These results would help clarify the observed differences in survival curves and add significance to the manuscript.
2. Despite of the observed sex differences, the authors only investigated the gene expression dynamics in female flies. It would be much more valuable to check whether the regulation of key aging-related and cell death-related genes differs between males and females, which could potentially account for the survival differences observed. Such an analysis, again, would significantly enhance the manuscript's impact.
3. Line 138, it states that “Expression of both PARGWT and PARGΔCter in a parg null mutant background successfully rescued the developmental arrest observed at the end of the third instar larval stage in parg mutant animals.” However, data supporting this statement is not provided.
Minor revisions:
4. Line 85: “We identified PARG binding to Act88F and TpnC47D loci (Fig.1B)” should be corrected to “(Fig. 1F).”
5. Figure 1E: It would be better to use blue for non-aging genes and orange for aging genes to maintain consistency with the coloring in Figure 1A.
Author Response
We want to express our gratitude for the invaluable comments from the reviewers and their constructive suggestions aimed at improving our manuscript. In response to their feedback, we have diligently revised and resubmitted our paper as an article, incorporating five primary figures, three supplementary tables, and two supplementary figures. It is essential to underline that this work represents a significant contribution to the field, presented here for the first time, with no prior publication.
Below we have provided a comprehensive response to the reviewer’s comments:
Reviewer 3:
The authors have made efforts to clarify the originality of their study, which builds on their previous work. The introduction and results sections have been improved, and unsupported claims have been removed. However, several issues and additional experiments are necessary to enhance the significance of the study and justify its publication.
R3: There is a noticeable difference in the survival curves between males and females expressing PARGWT and PARGΔCter. While various possibilities could explain this, such as differentially regulated pathways in stress responses, immune function, and metabolic processes, one key question is: Do the expression levels of PARGWT and PARGΔCter vary between males and females? This could lead to the different survival outcomes. Can the authors provide data on the expression levels of PARGWT and PARGΔCter in both male and female Drosophila? These results would help clarify the observed differences in survival curves and add significance to the manuscript.
Response: We thank Reviewer 3 for their insightful comments during these two rounds of revision that positively impacted our manuscript. The discrepancy in the survival between female and male flies expressing PARGWT and PARGΔCter is indeed interesting but falls outside the primary focus of our study, which aims to elucidate the regulatory mechanisms of PARG in the aging process. We believe that such an investigation would not substantially alter the conclusions or significance of our proposed model.
We have revised the discussion section to acknowledge the potential for sex-specific differences.
R3: Despite of the observed sex differences, the authors only investigated the gene expression dynamics in female flies. It would be much more valuable to check whether the regulation of key aging-related and cell death-related genes differs between males and females, which could potentially account for the survival differences observed. Such an analysis, again, would significantly enhance the manuscript's impact.
Response: We thank Reviewer 3 for this insightful comment. The observed discrepancy in survival between female and male flies expressing PARGWT and PARGΔCter is indeed interesting but falls outside the primary focus of our study, which aims to elucidate the regulatory mechanisms of PARG in the aging process. Investigating sex-specific gene expression dynamics would require extensive additional experiments, which are beyond the scope and timeframe of our current study. Moreover, these experiments will not significantly alter our proposed model.
We have revised the discussion section to acknowledge the potential for sex-specific differences.
R3: Line 138, it states that “Expression of both PARGWT and PARGΔCter in a parg null mutant background successfully rescued the developmental arrest observed at the end of the third instar larval stage in parg mutant animals.” However, data supporting this statement is not provided.
Response: We appreciate Reviewer 3 for their attention to this aspect. It's important to clarify that in flies, a parg null mutant is not viable; all animals progress normally until the end of the third instar larval stage but are unable to initiate pupation, thus precluding the emergence of adults. The version of PARGWT and PARGΔCter we used in this mansucript are expressed in a parg null mutant background in all the experiment presented Figure 2 and 3. Importantly, both variants successfully rescue parg mutant developmental arrest and generate adult flies which enables us to investigate the impact of the absence of the PARG regulatory domain on adult lifespan.
Minor revisions:
R3: Line 85: “We identified PARG binding to Act88F and TpnC47D loci (Fig.1B)” should be corrected to “(Fig. 1F).”
Response: We appreciate Reviewer 3 for bringing this typo to our attention. We have revised the results section to correct the error, now indicating the correct figure reference as "(Fig. 1F)." Thank you for your careful review, which contributes to the accuracy of our manuscript.
R3: Figure 1E: It would be better to use blue for non-aging genes and orange for aging genes to maintain consistency with the coloring in Figure 1A.
Response: We have adjusted the color scheme in Figure 1E to maintain consistency with the coloring used in Figure 1A-C, now using blue for non-aging genes and orange for aging genes. We appreciate Reviewer 3 for their valuable feedback, which has helped improve the clarity and coherence of our figures. Thank you for your continued engagement in refining our manuscript.
Round 3
Reviewer 3 Report
Comments and Suggestions for Authors
The authors recognize the importance but did not explore sex-specific gene expression dynamics due to time and scope constraints. This is understandable and does not entirely undermine its primary focus. Thus it may still hold value for publication. However, they should explicitly acknowledge in the discussion the limitation of not exploring the sex-specific effect in detail. This will provide transparency and set a clear direction for future research.
Author Response
Reviewer 3: The authors recognize the importance but did not explore sex-specific gene expression dynamics due to time and scope constraints. This is understandable and does not entirely undermine its primary focus. Thus it may still hold value for publication. However, they should explicitly acknowledge in the discussion the limitation of not exploring the sex-specific effect in detail. This will provide transparency and set a clear direction for future research.
Response: We thank Reviewer 3 for their thorough reviews and constructive feedback, which have significantly improved our manuscript. We have revised the discussion section to explicitly acknowledge this limitation:
"Furthermore, our findings reveal a more pronounced reduction in lifespan for female flies compared to male flies in the absence of the PARG regulatory domain. This compelling observation suggests potential sex-specific differences in the regulation of PARG protein. Although we did not explore these differences in the current manuscript due to time and scope constraints, they represent a promising avenue for future research. Investigating these sex-specific regulatory mechanisms could unveil critical insights and potentially transformative discoveries in the field of aging and molecular biology."
We believe this addition provides the necessary transparency and sets a clear direction for future research.